# Detecting Safety Anomalies in pHRI Activities via Force Myography

**DOI:** 10.3390/bioengineering10030326

**Published:** 2023-03-05

**Authors:** Umme Zakia, Carlo Menon

**Affiliations:** 1New York Institute of Technology, Vancouver Campus, Vancouver, BC V5M 4X5, Canada; 2Menrva Research Group, Schools of Mechatronic Systems Engineering and Engineering Science, Simon Fraser University, Burnaby, BC V5A 1S6, Canada; 3Biomedical and Mobile Health Technology Laboratory, Department of Health and Technology, ETH, Lengghalde 5, 8008 Zurich, Switzerland

**Keywords:** force myography technique, physical human–robot interactions, activities of daily life, pHRI-ADL, inlier and novel data detection, anomaly score, occupational safety monitoring

## Abstract

The potential application of using a wearable force myography (FMG) band for monitoring the occupational safety of a human participant working in collaboration with an industrial robot was studied. Regular physical human–robot interactions were considered as activities of daily life in pHRI (pHRI-ADL) to recognize human-intended motions during such interactions. The force myography technique was used to read volumetric changes in muscle movements while a human participant interacted with a robot. Data-driven models were used to observe human activities for useful insights. Using three unsupervised learning algorithms, isolation forest, one-class SVM, and Mahalanobis distance, models were trained to determine pHRI-ADL/regular, preset activities by learning the latent features’ distributions. The trained models were evaluated separately to recognize any unwanted interactions that differed from the normal activities, i.e., anomalies that were novel, inliers, or outliers to the normal distributions. The models were able to detect unusual, novel movements during a certain scenario that was considered an unsafe interaction. Once a safety hazard was detected, the control system generated a warning signal within seconds of the event. Hence, this study showed the viability of using FMG biofeedback to indicate risky interactions to prevent injuries, improve occupational health, and monitor safety in workplaces that require human participation.

## 1. Introduction

The use of industrial robots in manufacturing environments has relieved human workers from performing difficult and dangerous work. In recent years, robots working in collaboration with human workers in assembly lines have boosted productivity [1,2,3]. The assembly line robots are usually rigid-body and locally controlled with preset programs lacking advanced monitoring sensors. Hence, there are possible threats of injury for human workers while interacting with these heavy robots [4,5]. As a result, safety in human–robot interaction (HRI) is a growing concern in these environments. Though fatal accidents are rare in industrial plants, hand and wrist injuries are common in the production line. Observations reveal that common injuries for human workers occur in the wrists or hands (38%) due to exposure to immense mechanical forces from the robots [6,7,8]. Improper safety measures can lead to injuries during HRI. Therefore, safety measures and regulations are a priority in shared workspaces for surveillance and monitoring. A variety of safety measures, such as vision systems (cameras, image/depth sensors, tracking systems), ultrasonic/wideband/RF transceivers, capacitive sensors, or magnetic sensors for proximity detection, are implemented in the industrial environment [9,10,11,12]. Ensuring safety with these tools is still challenging due to their limited signal ranges and the difficulties of installation within a robot’s workspace, causing obstructions to signal among the robot, operator, and tools. 

Usually, a worker’s safety in industrial workspaces refers to a physical safety risk that mostly arises from collisions with robots. Hence, in most cases, maintaining a safe distance with robots would be appropriate. Recent research was conducted using a practical vision-based human safety system via mixed reality (MR) for establishing a minimum safe distance [13]. Ensuring the safety and optimizing the performance of the HRI task was investigated as a stochastic optimal control problem (OCP) that included sensory data on human motions. For collaborative tasks, adapting human behavior using a camera and optical tracker was taken into consideration for collision-free trajectory planning [14]. Real-time point-to-point motion planning in a shared workspace via a non-linear predictive model was studied in [15]. The authors in [16] investigated the recognition of human pose via sensors for context-awareness-based collision-free human–robot collaboration (HRC). Recent approaches used a supervisory control scheme often known as ‘shielding’ for safe robot operation in uncertain environments. The authors in [17] relied on predicting future shielding events that could override the robot’s nominal plan with a safety fallback strategy in critical situations. In this Industry 4.0 era, many of these studies used several monitoring devices and relied on deep learning approaches, IoT, cloud computing, and mixed reality for real-time control and safety. However, these studies focused on maintaining safe distances rather than direct contact between a human worker and a robot.

In industrial workplaces, unknown, abnormal human behavior or activities can create potential hazards around the sizable robots. To enhance the safety of human–machine interactions, an input signal from the human worker could provide insights about human intentions during physical human–robot interactions (pHRI). Monitoring human biosignals or biofeedback can enhance overall surveillance and compliance for the occupational health and safety (OHS) of the worker. Including physiological parameters of the human worker in the control system can improve safety protocols to avoid possible collisions, damages, or injuries. In a shared workspace with robots, collaborative tasks with a human worker often require dynamic changes or adjustments in the predefined tasks with robots. This requires implementing stable control systems for bidirectional human–robot interfaces [18]. To respond to the need for effective and safe human–robot collaboration, multi-modal symbiotic communication and control methods have been used in recent years. Methods such as voice commands, gesture recognition, haptic or tactile interaction, and brainwave perceptions are interpreted via machine learning and deep learning approaches to create awareness during interactions [19,20,21,22,23]. Along with the usual surveillances, these multimodal approaches can improve the efficiency of interaction and safety.

Among the variety of biosignals, electroencephalogram (EEG), electrocardiogram (ECG), electromyogram (EMG), and electrooculogram (EOG) are well-known examples. Among these, the non-invasive surface electromyography (sEMG) signal has been in practice for decades. This technique relies on detecting the electrical current of the underlying muscle bundles in action [24]. The sEMG technique has been studied extensively for understanding human intentions in collaborative interactions during pHRI [25,26,27,28,29,30,31,32,33,34,35,36,37,38,39,40,41]. These studies used various machine learning techniques (such as artificial neural network (ANN), 3D convolutional neural network (CNN), support vector regression (SVR), linear discriminant analysis (LDA), and K-nearest neighbor (KNN)) for estimating motion intentions. Recognizing human arm or hand gestures via inertial measurement unit (IMU) sensors and sEMG signals with audio and visual feedback was found to be effective in interactions with a collaborative robot (cobot) [42]. Cobots are becoming increasingly popular in shared workspaces, such as helping human workers in object transportation, helping nurses to lift objects, or helping doctors as intelligent diagnosis systems. These collaborative robots are designed to work with humans and can be programmed for safe interactions, made possible due to their built-in torque sensors in every joint. However, industrial robots differ from cobots as they do not have any built-in sensors to work safely around humans. They are essential in assembly and production lines, expensive, and not easily replaceable. Thus, ensuring human workers’ safety in such industrial environments is important.

In recent years, a contemporary force myography (FMG) technique has gained research interest, like the sEMG technique. FMG is a non-invasive, wearable technology that usually employs force sensing resistors (FSRs) to detect resistance changes when the pressure changes during muscle contractions [43]. As a promising biotechnology, the FMG technique was studied in human–machine interaction (HMI) and human–robot interaction (HRI) [44,45,46,47]. Common industrial tasks such as object transportation or handover may require the worker to apply hand forces during interactions with the robot. Estimating the applied hand force to interact with a robot in dynamic motions using FMG signals was found favorable in physical interactions [48,49,50,51,52,53]. These studies implemented compliant collaboration, where a robot’s trajectory was dictated by the applied hand force of a human worker. Admittance control was established for the compliant collaboration, where the robot followed the same path as the human’s applied force and direction. Supposedly, the use of FMG biosignals could help to ensure safe interaction during collaborative tasks by signaling the robot to push away if it approached the human worker in an unsafe manner. However, when a robot approaches a human worker at a faster speed, it might become difficult to signal the robot to stop. Therefore, establishing additional safety mechanisms via FMG signals will be more practical.

In this study, our objective was to identify avoidable potential safety threats that could cause plausible injuries during direct, physical interactions between a human worker and a robot. The proposed safety mechanism was investigated as an additional protection layer with an underlying compliant collaboration control layer (admittance control) using FMG biosignals. In the industrial workplace, interactive tasks between human workers and robots are preset and are repeated daily. We defined such regular interactive tasks as ‘pHRI-ADL’ (activities of daily life in pHRI) and use this term hereinafter in the rest of this article. Any unwanted interaction outside the scope of pHRI-ADL activities was considered an unsafe situation. Considering this, a few daily interactive tasks between a human participant and a robot (a serial manipulator in this case) were considered as normal activities, and data were collected using a wearable FMG forearm band. Using unsupervised classification techniques, a model was trained with regular pHRI-ADL data to learn the latent feature distributions. Three separate unsupervised data-driven approaches, namely the isolation forest (iForest), one-class support vector machine (OCSVM), and Mahalanobis distance (MD) algorithms, were implemented in this study. These trained models were evaluated separately for identifying unwanted interactions where the pHRI activity was different than usual. The unsupervised trained models detected inliers or novel data compared to the normal data, i.e., they detected unsafe, unwanted interactions quickly and generated alerts to warn about the hazardous situation. Figure 1 shows the proposed occupational health and safety monitoring system using the FMG techniques. The main contributions of this study were as follows:−proposing an extra layer of occupational safety feature for human–robot workplaces in addition to the compliant collaboration control layer;−introducing single-mode biosensory FMG data for both control and safety;−defining normal, daily interactions between a human worker and a robot that were preset and carried out regularly as ‘pHRI-ADL’ activities via FMG-based compliant collaboration;−implementing unsupervised learning techniques to understand the daily human–robot interactive tasks via iForest, OCSVM, and MD algorithms by training models with the normal pHRI-ADL distributions;−evaluating the trained models in detecting unsafe interactions that were novel, inlier, or outlier to the normal distribution;−generating warnings when unsafe or hazardous interactions were detected;−the feasibility of the proposed system to react quickly in generating alerts for manual or automated system override to prevent injuries.

In this Industry 4.0 era, automobile industries are swiftly embracing technologies such as IoT, cloud computing, artificial intelligence, and big data analytics to achieve increasing productivity. For collaborative tasks, the FMG technique can support controlled interactions, i.e., compliant collaboration via data-driven approaches. In addition to this, this study investigated the viability of FMG biofeedback to support the occupational health and safety compliance of a human worker in a robot’s workspace via Industry 4.0 technologies.

The rest of the article is organized in several sections. Section 2 describes the materials and methods, while Section 3 describes the obtained results and performance evaluations of the proposed framework. Section 4 discusses limitations, related studies, and future work, while Section 5 concludes this article.

## 2. Materials and Methods

### 2.1. UnSafe pHRI Detection: Anomoaly or Novelty?

Safety detection in human–robot interaction has become crucial due to the increasing human–robot collaboration in industrial workplaces. Physical human–robot interactions are unpredictable and dynamic in nature, and there are infinite ways in which a human might act abnormally. Datasets derived from normal and abnormal pHRI activities will differ from each other, providing a basis for determining unsafe or unwanted pHRI activities that happen suddenly and are different from usual daily pHRI activities. Unsafe/unwanted activities are usually an unsupervised learning task problem that attempts to detect anomalies or novelties in comparison to the normal data instances. However, anomalies and novelties are both closely related to outliers and can be used as a safety indicator for monitoring abnormal behavior. Both anomaly and novelty detection rely on defining what comprises a “likely or normal” profile. An anomaly can be seen as an unexpected, sudden change that deviates from the normal profile. Thus, we consider anomalies as unusual or unexpected data instances within a dataset. Likewise, novelties are also anomalies in data, meaning unusual data that are new and do not occur regularly or are simply different from the others. Therefore, we can employ novelty detection for rare events, of which we have very few samples, because these data exist only in new instances and do not exist in the normal dataset. Thus, outlier detection focuses on detecting anomalies in training data, whereas novelty detection helps in detecting anomalies in new data with uncontaminated training data. The novelty detection method helps in determining if these new data are within the norm (inlier) or outside of it (outlier). Our focus is novelty detection because unexpected pHRI interactions do not happen regularly; these data are completely new or unknown and may overlap with inliers within the normal distributions. Therefore, in this study, we considered usual pHRI activities of daily life (pHRI-ADL) as the normal dataset, while novelties were defined as unsafe data from unwanted pHRI activities.

### 2.2. Unsupervised Learning Algorithms

Unsupervised learning is designed to characterize training data that have no true labels associated with them. This type of algorithm uses data points (covariates or the features) to discover useful patterns. On the other hand, supervised learning algorithms rely on labeled training data points for training the models. In many real-life situations, most of the time, it is difficult to obtain labels, or sometimes incomplete or partial labels are collected. For a large dataset, it becomes time-consuming or difficult to annotate labels. Instead of specifying labels, learning the latent patterns can provide us with intuitive knowledge through unsupervised learning algorithms. Finding anomalies or novelties in the data is an important aspect in monitoring fault identification or malicious or unwanted activity and can be used to determine occupational safety hazards to avoid fatal injuries in the workplace.

To detect unwanted or unsafe pHRI activities, in this study, we relied on three unsupervised algorithms, namely isolation forest (iForest), one-class support vector machine (OCSVM), and Mahalanobis distance (MD). The unlabeled FMG signals of several, distinct pHRI-ADL data instances were used as a normal dataset to train a corresponding model. The model was then evaluated for detecting unknown situations deemed as an unsafe interaction. These algorithms are described in brief below.

(a) iForest algorithm

The isolation forest algorithm calculates the distance of a data point from the rest of the data points and isolates it. It works well for anomaly detection in big data with linear time complexity and low memory requirements. In this algorithm, every single isolation tree is trained for a subset of training observations, sampled without replacements. An isolation tree is grown by selecting a split variable and split position randomly for a subset until each observation is set in a separate leaf node [54]. Since anomalies are few, these samples land in separate leaf nodes with shorter distances from the root node than the normal points.

(b) One-class learning or unsupervised SVM (OCSVM) algorithm

One-class support vector machine (OCSVM) is an unsupervised machine learning method using a Gaussian kernel function. It learns from one class of data and then classifies any new data by separating the new data from the original data [55]. This algorithm standardizes the input data and selects an appropriate kernel scale parameter using a heuristic procedure.

(c) Mahalanobis distance (MD) algorithm

For multivariate data analysis, Mahalanobis distance is used to determine the distance between two different distributions (training and test data). As the squared MD follows a chi-square (χ^2^) distribution from samples to distribution, this distance can be used to identify amnomalies based on the critical values of the χ^2^ distribution. It is used instead of the Euclidean distance to handle skewed data by considering the covariances of the distributions. It converts the distributions to standard normal distributions with uncorrelated variables by changing variances into unit variances and then measures distance as in the Euclidean distance [56].

### 2.3. Physical Human–Robot Interaction Platform

For anomaly or novelty detection, interactions between a human participant and a Kuka robotic arm (KUKA LBR IIWA 14 R820, KUKA Robotics, Augsburg, Germany) were observed. The study protocol was approved by the Office of Research Ethics at Simon Fraser University, Canada (approval code: 20200226) and the participant gave his informed written consent. A 16-channel FMG forearm band was custom-made using 16 force sensing resistors (FSRs) whose resistances changed when pressure was applied on their surfaces, as shown in Figure 2a. The forearm FMG band had 16 FSRs (TPE 502C, Tangio Printed Electronics, Vancouver, BC, Canada), and it was roughly 30 cm long and connected to an external PC (Intel Core i7 processor and Nvidia GTX-1080 GPU) via a data acquisition device (NI USB 6341, National Instruments, Austin, TX, USA) to collect data at ~50 Hz. The band was worn on the participant’s dominant right arm, covering the forearm muscle belly, to collect muscle contraction readings during intended interactions. The captured instantaneous FMG signals were mapped to applied forces in dynamic motion via a data-driven approach.

The target environment of our work was the industrial manufacturing assembly line, where huge mechanical robots were engaged. These robots had no torque sensors in their joints to recognize externally applied forces by human workers during interactions. Replacing these robots or including joint torque sensors for safe human–robot collaboration was neither cost-effective nor feasible. As we needed to investigate improved safety and surveillance techniques for such a workspace, we monitored human interactions with a Kuka robot in this research. The manipulator had an 820 mm reach with a 14 kg payload and had built-in torque sensors in all joints except the end-effector. However, we did not utilize these torque sensors, so that we could replicate similar functionalities of the industrial robots. It came with its own Kuka Sunrise Controller and safety measures, such as an emergency stop button. At the flange, a customized cylindrical shape gripper as the end-effector (EEF) was attached for grasping and ease of interaction. Externally applied hand force was recognized by an external 6-axis FT sensor (loadcell) connected between the flange and the gripper. Figure 2b shows the Kuka robotic arm with the customized EEF oriented at a fixed angle {0, pi, 0}. The Sunrise toolbox and V-Rep robot simulator from Coppelia Robotics were used to establish data communication and control between an external PC (Intel Core i7 processor and Nvidia GTX-1080 GPU) and the Kuka Sunrise controller. Admittance control was established for compliant collaboration and codes were written in MATLAB. For ease of interaction, motion trajectories were confined in flexible 6-axis rectangular areas rather than fixed paths. For admittance control, the 6-axis FT sensor was attached externally to the robot as a true force label generator for training a convolutional neural network (CNN) model to predict the applied hand force from the instantaneous FMG signals. The desired joint angles were read from the V-Rep simulator using inverse kinematics (IK) and was sent to the Kuka Sunrise Controller. Then, the current position and orientations of the EEF were collected from the Kuka Sunrise Controller, and the desired EEF position was generated from the predicted applied hand force on the EEF and its current position. This control mechanism could be implemented to control and monitor large industrial assembly line robots that do not have any joint torque sensors to recognize the external environment. Further description of this pHRI platform can be found in [51].

### 2.4. pHRI Data Collection

The pHRI datasets were collected during dynamic interactions between a participant and the Kuka robot in 1D, 2D, and 3D space. The participant grasped the cylindrical gripper in a standing position in front of the robot and applied forces in dynamic motions wearing a 16-channel FMG forearm band on his dominant right arm, as shown in Figure 2c. Data for several interactions were collected in the 1D, 2D, and 3D planes. For all interactions, 5 repetitions of sample data (applied human forces in dynamic motions during interactions with the robot) were collected. All data were balanced, and no missing values were present in the datasets. Thus, preprocessing was not required to reduce memory consumption and speed up training. Raw FMG signals were used as the predictor data, organized as CSV files. Each row corresponded to one observation, while each column corresponded to one predictor variable. Hence, for the 16-channel FMG band, 16 features or predictor variables were present. Each interaction continued for around 60–90 s, termed as one ‘repetition’, and contributed approximately 600–900 × 16 samples at a ~10–12 Hz sampling rate. A detailed description of this dataset can be found in [53]. Table 1 shows the normal and abnormal datasets that were used in this study for anomaly detection in human–robot interactions.

▪Normal Scenario: pHRI-ADL Training Dataset

The preplanned human–robot interactions that happened between the human participant and the Kuka robot as regular tasks were termed as the activities of daily life in pHRI (pHRI-ADL). These activities included interactions between the participant and the Kuka robot in 1D (X, Y, and Z directions), 2D (XY and YZ planes), and 3D (XYZ plane). In each task (such as interactions in 1D-X), the human applied an external force in a distinguished direction, and the robot’s trajectory became different from other interactions or tasks. Five (5) repetitions of similar FMG data were collected for one task or interaction. Hence, a total of 60,743 × 16 FMG samples from 30 repetitions (1D plane: 15 reps, 2D plane: 10 reps, 3D plane: 5 reps) were collected. The variety of FMG data formed a normal training dataset to train a few selected unsupervised models. 

▪Unsafe Scenario: Target Dataset

For novelty detection, interaction in the 2D-XZ plane was considered as anomalous data, meaning that this interaction was unwanted and unsafe. Five repetitions of data, approximately of ~680:950 × 16 samples, were collected for cross-trial evaluation. These test data were collected when unwanted interactions between the participant and the robot occurred as an unplanned task. Each repetition was around one minute, and, for research purposes, the participant repeated the same interactions several times. Thus, the suspicious activities happened cyclically.

Interactions between the participant and the Kuka robot in the 2D-XZ plane were considered as anomalous data. During this unsafe and unwanted interaction, the participant, standing in front of the Kuka robot, grasped the cylindrical gripper and applied forces in the 2D- ‘XZ’ plane. In Figure 3a, the purple shaded box from the V-Rep simulator shows the area where interactions took place. Figure 3b shows the pHRI interaction, where the participant first pushed the robot inwards in the X direction, pushed up in the Z direction, and then pulled outwards in the X direction and finally pulled down in the Z direction. For ease of interaction, the motion path was confined to a 6-axis rectangular area. This 2D-XZ pHRI activity was never performed as a regular activity; however, it had similar interactions to the 1D-X and 1D-Z embedded in the data—for example, when the participant pushed the Kuka robot inwards in 1D-X by approximately 200 mm, or pulled outward in 1D-X by approximately 200 mm.

Figure 4 shows the FMG samples of pHRI training and test data used in this study. The training data had a variety of interactions in the 1D, 2D, and 3D planes. Hence, it was not possible to know the exact interaction category of the plotted training FMG signals.

## 2.5. Implementing Novelty Detection via Unsupervised ML Algorithms

To detect anomalies during unsafe pHRI activities in the abnormal scenario, several unsupervised models were trained with uncontaminated (without outliers) training data (pHRI-ADL dataset) and evaluated separately. These scripts were written in MATLAB.

▪
*iForest algorithm*


In this algorithm, isolation forest was created with regular training data (pHRI-ADL) that had no contamination or outliers. It treated all training observations as normal observations by default. To detect anomalies in the new, abnormal data (test data), the average path length to reach a leaf node from the root node in the trained isolation forest was calculated for each observation, and an anomaly indicator and a score were returned. During evaluations, the algorithm identified observations with anomaly scores above a score threshold (was set during training) as anomalies. Anomaly scores were computed for each row of an observation. A value of logical 1 (true) indicated that the corresponding row of the input data was an anomaly.

▪
*OCSVM algorithm*


The OCSVM algorithm treated all training observations as normal. Input data were standardized, a radial basis function (RBF) kernel was used, and an appropriate kernel scale parameter was selected using a heuristic procedure. A model was trained with uncontaminated training data of pHRI-ADL (data with no outliers) and was evaluated in detecting anomalies in new, abnormal test data by returning anomaly indicators and scores for the new data. By default, it identified observations with scores above a threshold value as anomalies.

▪
*Mahalanobis Distance*


In this multivariate data analysis, we used Mahalanobis distance to determine the distance between the two different distributions, pHRI-ADL data and abnormal data. By default, the algorithm assumed a multivariate normal distribution with no outliers in the training data based on the critical values of the chi-square distribution. First, Mahalanobis distances were computed along with the mean (μ) and covariance matrix (∑) for the training (pHRI-ADL) data. Then, the estimated μ and ∑ were used to compute distances for the test data.

## 2.6. Performance Analysis Tools

▪
*Anomaly Score*


An anomaly can be defined as the variation between an actual value and a long-term average value. Any data points outside the range or new within the range of the normal distribution are marked as anomalies. The anomaly score is a value from 0 to 1, where values less than 0.5 are considered normal and values above 0.5 are anomalies. It can also be set as a negative score value with a large magnitude indicating a normal observation, and a large positive value indicating an anomaly. Anomaly scores have the same length as the observation, and each element of the scores contains an anomaly score for each observation, i.e., one score for the corresponding row of data. In this complex multivariate (16-channel feature distributions) anomaly detection, the outlier/inlier is usually a joint uncommon score on at least two variables.

For the *iForest algorithm* [54], outliers were identified using anomaly scores based on the average path lengths over all isolation trees. The algorithm computed the anomaly score *S_iF_*(*x*) of an observation x by normalizing the path length *l*(*x*):(1)$$s_{iF}\left(x\right)=2^{-\frac{E\left[l\left(x\right)\right]}{C\left(n\right)}}$$
where *E*[*l*(*x*)] was the average path length over all isolation trees, and *C*(*n*) was the average path length of n observations. The score approached 1 when *E*[*l*(*x*)] approached 0. Therefore, a score value close to 1 indicated an anomaly. The score approached 0 when *E*[*l*(*x*)] approached n − 1. Moreover, the score approached 0.5 when *E*[*l*(*x*)] approached *c*(*n*). Therefore, a score value smaller than 0.5 and close to 0 indicated a normal point. A threshold value was used to identify observations in the new data as anomalies. The default threshold of the maximum score value was determined when the isolation forest was trained with the regular pHRI-ADL dataset. By default, the algorithm identified observations with scores above the threshold as anomalies.

For the *OCSVM algorithm* [55]*,* a numeric anomaly score in the range (−∞, ∞) for each observation was returned. A negative score value with a large magnitude indicated a normal observation, and a large positive value indicated an anomaly. As with iForest, a threshold value was used to identify observations in the test data as anomalies. The default threshold of the maximum score value was determined when the OCSVM model was trained with the pHRI-ADL dataset. By default, the algorithm identified observations with scores above the threshold as anomalies.

For the *Mahalanobis distance algorithm* [56], the distance from a vector *x* to a distribution with mean *M* and covariance *C* was calculated as
(2)$$Md=\sqrt{\left(x-M\right)C^{-1}{\left(x-M\right)}^{\prime }}$$

Here, the distance *Md* represented how far *x* was from the mean in the number of standard deviations, *x* was the vector of the observation (row in a dataset), *M* was the vector of mean values of the independent variables (mean of each column), and *C^−^*^1^ was the inverse covariance matrix of the independent variables. For anomaly detection, MD between every test data point and center in the n-dimension training data (pHRI-ADL dataset) was calculated. For novelty data evaluation, the obtained mean and covariance matrix during training were used in computing the Mahalanobis distance between the test (abnormal) data and the normal distribution, and outliers were found based on these distances. The maximum MD score value from the training was used as the threshold value and observations with scores above the threshold were identified as anomalies.

▪
*Histogram-Based Novelty Score*


The performance of the algorithms in detecting patterns or anomalies was plotted and visualized in histograms. The training distributions and test data points in the plotted histograms showed quick identification of outliers, patterns, or trends based on the degree of anomaly calculations. In this multivariate anomaly detection, a single histogram for each single feature was calculated, scored separately, and combined at the end.

▪
*Laplacian Score*


This unsupervised linear feature extraction method computed the Laplacian score for each feature based on the locality preserving property (data from the same class were often close to each other). It selected variables with the smallest scores and ranked the features of the observations using the Laplacian scores. This function selected two important features, i.e., the prominent FMG signal readings from two FSR sensors that influenced the anomaly detection most.

## 3. Results

### 3.1. Performance of the Three Algorithms

The three unsupervised classifiers were pretrained with the preset, regular activities, i.e., with the pHRI-ADL data, and the trained models were saved as mat files for future monitoring. These were separately evaluated on novel pHRI test data that were considered unsafe for the human worker. Cross-trial evaluations were carried out for all five repetitions (rep1–rep5) of the test data. The cross-trial tests replicated a real-world situation where test data were unpredictable or unprecedented, even during repetitions of the same task. A histogram of the score values *s* of training data (in blue) and *s_test* of test/new data (in brown) was plotted with a vertical line at the score threshold, as shown in Figure 5. These histogram plots showed that the test anomaly scores were distinguishable compared to the training scores. This justified the viability of the proposed system even when the test data were inliers (overlapping with normal data).

For cross-trial evaluations of the test data (rep1–rep5) via a selected algorithm, the pretrained model returned the predicted anomaly scores of the test data. For each observation of data, a score was generated, i.e., for repetition N of the test data (M X P samples), M scores between 0 and 1 were obtained. Test anomaly scores higher than 0.45 were considered as novel data (score was set to 1, indicating abnormal data) while lower scores were considered as regular data (score was set to 0, indicating normal data). A confusion matrix was generated for each test repetition, as shown in Table 2. The actual test data were considered to have scores of 1 (only abnormal data were present, i.e., negative/TN = 0, negative data/normal were absent). The predicted scores were either 1 or 0 (either abnormal or normal). The confusion matrix indicated predicted test scores either as abnormal (true positive/TP) or normal (false negative/FN) data. The set threshold helped us to avoid false positive (FP = 0) alarms and true negative (TN = 0) because the real test data lacked normal or negative data.

For performance evaluations, accuracy in detecting novelties was calculated to find the ratio of correctly identified abnormal test data compared to the total test data. We defined accuracy in detecting novelties using the following equations:(3)Acc in detecting abnormality=(TP+TN)/(TP+FP+FN+TN)
(4)Acc in detecting normality=1−Acc in detecting abnormality

The results obtained using Equations (3) and (4) are included in Table 3, Table 4 and Table 5 under the columns ‘Novelty Detection’ and ‘Misclassified as Normal’.

▪
*Detecting Anomalies via iForest algorithm*


The pretrained iForest model was able to detect novelties during unsafe abnormal pHRI activities in 2D-XZ in each interaction/repetition with accuracies of 73~96%. In each repetition, the model quickly predicted the abnormality in samples in less than 2 s. For example, for each row of observations in repetition 1 (830 × 16 samples), one score was generated, i.e., 830 anomaly scores were predicted, where 92% of the test data were detected as novel and unsafe in a total elapsed time of 1.52 s. Lower elapsed times were observed for the other repetitions. In repetition 3, the model was approximately 73% accurate in detecting abnormalities, which was lower than the other repetitions.

▪
*Detecting Anomalies via the OCSVM algorithm*


The pretrained OCSVM model detected novelties in the test data, i.e., in each interaction/repetition of the unwanted pHRI activities, with accuracies in the range of 8–25%. The model quickly predicted the abnormality in each sample of the repetitions in less than 1.9 s. For example, in repetition 1 (830 × 16 samples), 830 anomaly scores were predicted with the highest accuracy of 25% in unsafe pHRI detection within a total elapsed time of 1.2 s. Lower elapsed times were observed for the other repetitions. In repetition 3, this model could only detect abnormal data in 9% of the test sample data.


*Detecting Anomalies via the Mahalanobis Distance algorithm*


The pretrained iForest model was able to detect novelties during unsafe abnormal pHRI activities in the 2D-XZ plane in each interaction/repetition with accuracies of 100%. In each repetition, the model quickly predicted the abnormality in samples in less than 1.3 s. For example, for each row of observations in repetition 1 (830 × 16 samples), one score was generated, i.e., 830 anomaly scores were predicted with 100% accuracy, where the test data were detected as novel and unsafe in a total elapsed time of 1.04 s.

The Mahalanobis and the iForest models could identify unsafe pHRI activity better than the OCSVM model using the same test data by identifying more data as anomalies.

### 3.2. Reaction Time for Safety Protocol Initialization

Once a model could detect abnormal pHRI activity, the next step was to create an alert signal to indicate a safety breach for the human worker. This could trigger an emergency safety protocol (either manually or automated) to initiate and prevent any severe injuries. To measure the appropriateness of the proposed system, a detailed analysis was carried out on the reaction time of abnormality detection to alert generation, as shown in Figure 6 and Table 6. To understand the impact, we analyzed the worst-case diagnosis of the iForest algorithm in evaluating test data (rep 3: 686 × 16 samples). In this repetition of test data, the iForest model could detect fewer test samples as abnormal data (189 × 16 samples) than other test repetitions. With an average sampling frequency of 10 Hz, each row of observations was of 0.1 s or 100 ms duration, i.e., one anomaly score was created in every 100 ms. For 686 × 16 samples of unsafe pHRI interactions (rep 3 duration), the algorithm took 3.9 s from loading the pretrained model to generating anomaly scores over the whole observation.

It was observed that the model detected 189 × 16 samples as anomalies with scores higher than 0.45. In this case, the elapsed time from detecting the unsafe pHRI activity to generating an alert signal was approximately 1.036 s. Obviously, the model was faster to react and respond even when minimal samples were detected as abnormal activities.

### 3.3. Normal vs. Abnormal or Unsafe pHRI Activity Detection 

To understand the performance of the algorithms, first, the training data were observed to identify if anomalies were present (Figure 7). Then, the training data were contaminated with the test data, as shown in Table 7. Finally, the contaminated training data were again evaluated to detect any possible anomalies (Figure 8).

Figure 7 shows the pHRI-ADL training data distributions of the normal points in the 1D, 2D, and 3D planes. The graphs generated for the three algorithms were almost similar for the same training dataset. The *Laplacian function* was used to find the highest-ranked features among the 16-channel FMG signals. To visualize the detected outliers, data dimensions were reduced.

In Figure 8, we contaminated pHRI-ADL training data with the unsafe test data from interactions in the 2D-XZ plane. The graphs in this figure show the presence of novel data (red points) in the contaminated training dataset. Features 14 and 15 ranked highest among the 16-channel FMG signals. To visualize the detected outliers, the data dimension was reduced.

These results support the justification of novelty detection instead of anomaly detection. As the test data were inliers to the normal pHRI-ADL distributions, these were all over the normal distributions rather than appearing as outliers, and hence anomaly detection could not find these inliers. The fractions of detected anomalies in the contaminated training data by the three unsupervised algorithms were 0.0506 (iForest), 0.0507 (OCSVM), and 0.0507 (MD). The novelties identified by these methods were located near each other in the reduced dimension. The computed fraction of data for which the three methods returned the same identifiers was 0.0241. This showed that identifying sudden unwanted interactions occurring during the normal HRI activities was possible.

## 4. Discussion

### 4.1. Discussion and Limitations

To evaluate the performance of the classifiers in detecting anomalies, we used an anomaly score, histogram-based novelty score, and Laplacian score with the same baseline parameters and thresholds for consistencies. For example, for all three algorithms, an anomaly score of less than 0.5 was considered normal and values above 0.5 were considered anomalies. The detection of anomalies was plotted and visualized in histogram-based novelty scores. The training distributions and test data points in the plotted histograms rapidly showed the identification of outliers, patterns, or trends based on the degree of anomaly calculations. The Laplacian score was computed for each feature based on the locality preserving property (data from the same class were often close to each other). It selected variables with the smallest scores and ranked the features of the observations using the Laplacian scores.

For compliant collaboration during training and testing, motion trajectories were not fixed between a start and end point; rather, it was bounded by 6-axis rectangular areas in the 1D, 2D, and 3D workspaces [51]. This allowed flexible dynamic motions within certain ranges, instead of restricted movements. To evaluate the classifiers, we trained all three models with the same training datasets that were collected during the preset, daily human–robot interactions. Each training interaction in the 1D-X, Y, Z plane, 2D-XY, YZ plane, and 3D-XYZ plane was different and, hence, the collected data were different in terms of the applied interactive forces and motion trajectories. Even in a selected interaction, such as in the 1D-X plane, the five repetitions of collected data were not exactly same. These pretrained models were evaluated on test data where unwanted interactions happened between the human and the robot. The anomaly score, histogram-based novelty score, and the Laplacian score were calculated for each model. For fair comparison, cross-trial evaluations of the detection of unwanted interactions were carried out several times. Likewise, due to uncertainties involved, the five repetitions of the test interactions were not exactly same. Thus, the cross-trial evaluations showed the viability of the proposed system even for a single test repetition scenario.

As this study was done in a strictly monitored experimental environment with a certain setup, we defined a few regular preset tasks and one unwanted test scenario where a human participant interacted with a 7-DoF serial manipulator. The models were trained with different interactions in the 1D, 2D, and 3D planes, which were separate from each other. This was done with the intention to represent a variety of pre-defined industrial human–robot interactive tasks as regular tasks. In the test scenario, we selected an interaction that was unwanted, undefined, impulsive, and suddenly occurring between the human and the robot. The models were examined separately five times as cross-trial evaluations of the test scenario. It was observed that the models were able to detect anomalies or novel, unsafe interactions and could detect the occurrence of a potential health and safety hazard within a second, not only for the best case (100% anomaly detection via MD model) but also for the worst case (9% of anomaly detection via OCSVM model).

This study relied on data-driven models to learn the latent features of FMG biosignals in detecting anomalies or safety hazards. The trained model used a variety of pHRI-ADL or regular data to detect unusual or irregular pHRI data. During evaluation, detecting novel or unwanted interactions was possible because the hazardous movement was significantly different from the usual tasks. Hence, for real-time practical applications, the collection of every possible regular activity’s data is required to ensure safety in industrial workspaces. In a situation wherein safety hazards arise during usual interactions and the worker tries to continue the preset fixed task, it might become difficult for the data-driven model to detect anomalies. The model might not succeed to determine safety hazards if the unwanted interaction becomes similar to the regular data. Therefore, the proposed safety detection in this study can be implemented as an additional feature with the usual surveillance monitoring that prevails in the industry.

In contrast to multimodal safety systems, we relied on one type of biosensory data, i.e., the FMG data only, for both the underlying compliant collaboration control layer and the proposed safety monitoring as an additional layer. This study was an extension of the previous works where instantaneous applied hand forces were predicted for admittance control via a supervised convolutional neural network model [51]. The focus was to investigate safety in collaborative tasks when the human worker had direct contact with the robot, rather than implementing a collision-free or collision avoidance system. With the FMG-based admittance control, a human worker could manually push away the robot if needed, but it was not possible to know the occurrences of potential safety hazards. This motivated us to build an additional safety layer for unwanted pHRI detection with the same biosensory data. This helped in reducing control signal communication and computational complexity by having one type of sensory data instead of multimodal data. Hence, the proposed system can be used as a single source of sensory data in combination with prominent surveillance systems such as the vision system. It can provide an additional layer of protection within the existing safety protocol for human–robot workspaces, rather than as a stand-alone safety system. In the future, FMG biosignals can be used with other sensory devices such as IMU data or with vision systems for a comprehensive workspace safety design. Therefore, although the proposed system was found viable with the experimental training and test applications, we believe that it could feasibly be applied in industrial scenarios of real human–robot interactive applications, although further studies involving more human participants are required in the future.

### 4.2. Related Studies, Application, and Future Work

Due to increasing human robot collaborations, many safety monitoring systems prevail in the industry and ongoing research is carried out focusing on anomaly detection. Among all these systems, video anomaly detection provides an extra layer of protection in addition to the robot’s own safety settings, and only uses the original surveillance videos, with no extra cost [57]. Moreover, many smart sensory systems have been developed for monitoring HRI activities. The authors in [58] used a capacitive sensory system to monitor data quality using the OCSVM algorithm to achieve safe HRI. Abnormalities can also be used as an indicator of health-related challenges. In one study, an ensemble of novelty detection models consisting of iForest, OCSVM, local outlier factor (LoF), and robust covariance estimation (RCE) was used for anomaly detection [59]. This study was conducted to detect abnormalities in the daily routines of older people. In another study, upper limb spasticity was observed via the iForest algorithm by detecting spikes in the surface electromyography (sEMG) of the biceps and muscle resistive force [60].

In this study, observing human–robot interactions via FMG biosignals provided a method to detect unsafe human activities. In our previous studies [48,49,50,51,52,53], human intentions during motions and applied interactive forces for pHRI activities were successfully incorporated in effective control system designs. In addition to the compliant collaboration, this proposed unwanted pHRI activity detection system will provide an extra layer of protection and will improve occupational health and safety monitoring. Moreover, it can be a feasible solution for injury prevention in safer robotic systems designed for humans. Nevertheless, more studies are required to investigate different hazardous scenarios where the test data might be difficult to differentiate. As in this study, the training data had normal data only; there were no abnormal data included. The test data had unwanted, abnormal data, so there were no normal data included. However, even during unwanted interactions in the 2D-XZ plane, some movements might have resembled some normal activities (such as 1D-X, 1D-Z).

Adopting data-driven approaches was found to be advantageous in facilitating collaboration by learning motion patterns to recognize human feedback. These approaches also helped to avoid the complex, time-consuming dynamic arm modeling of the individual human participant. The unsupervised learning techniques used in this study showed the viability of detecting normal or abnormal movements during human–robot interactions and generated alert signals for manual override. Further improvements can be achieved by developing an automated decision-making system with collision avoidance trajectory planning for the robot once unsafe interactions are identified.

In most pHRI studies, monitoring workers’ safety is based on camera surveillance. To our knowledge, there is no other pHRI study using FMG biosignals to detect occupational safety hazards in robotic workspaces. In the future, this work may be compared with related biosignal-based human–robot interaction studies. Further studies can be conducted involving more participants and performing industrial collaborative tasks with robots. In this preliminary study, our focus was to investigate the viability of a single type of biosensory data for safety monitoring along with compliant collaboration via admittance control. The FMG signals provided useful information to interpret human motions and to detect possible threats or hazardous situations. The study was conducted in a strictly monitored experimental environment where one human participant wearing a forearm FMG band interacted with a serial manipulator. Future studies can include another FMG band on the upper arm to record shoulder abduction and adduction, an IMU sensor to track the arm pose and orientation, and a vision system to capture unsafe interactions. Multimodal sensory data would be more reliable in understanding human intentions and monitoring potential safety threats during interactions in industrial workspaces. Moreover, the approaches adopted here can be further studied in other fields where understanding human activities can be insightful.

## 5. Conclusions

Common industrial human–robot collaborative tasks are usually preset daily activities for a human worker to interact with machines. Hence, observing human intentions and motions during regular interactions via biosignals can improve occupational health and safety monitoring. In this study, regular pHRI activity data were used to train a few machine learning models and were interpreted as activities of daily life in pHRI (pHRI-ADL). Observations on daily pHRI activities provided us intuitive information on human intentions, motions, and interactive applied forces by diagnosing the underlying signal patterns and learning latent feature distributions. Any unwanted interactions were diagnosed as an unsafe situation to prevent injuries by implementing safety measures in the workplace. Three unsupervised learning algorithms (isolation forest, one-class SVM, and Mahalanobis distance) were used to understand regular activities and then were evaluated in detecting unwanted pHRI scenarios. These algorithms could detect novelty or unsafe interactive activities quickly, even when test data were inliers, i.e., where the unknown FMG signal distributions were within the normal distributions. The proposed models were able generate anomaly scores every 100 ms, detected situations wherein safety concerns arose, and created alert signals within 1 s of the occurrence. Hence, this study showed that FMG biofeedback can be used to indicate unsafe situations to prevent injuries, improve occupational health, and monitor safety via unsupervised machine learning algorithms. With the advent of the IoT era for body-wearable devices [61], the wearable FMG armband can be a viable solution as a control mechanism and be part of a safety monitoring system along with other sensory devices.

## Figures and Tables

**Figure 1 bioengineering-10-00326-f001:**
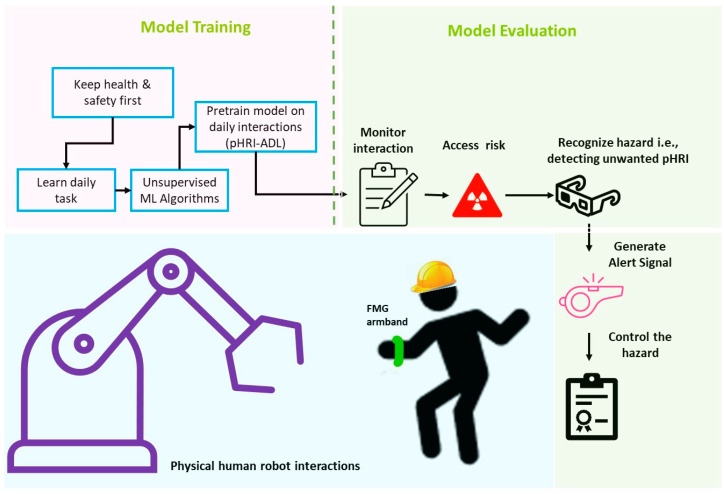
The proposed FMG-based occupational health and safety monitoring system.

**Figure 2 bioengineering-10-00326-f002:**
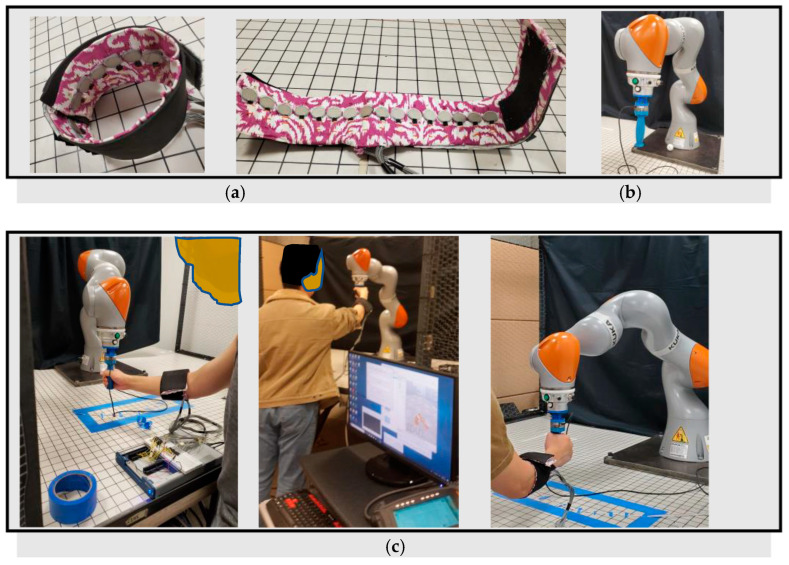
The pHRI setup: (**a**) 16-channel FMG forearm band, (**b**) Kuka robotic arm with cylindrical gripper, (**c**) regular, preset activities between a human participant and the Kuka robotic arm.

**Figure 3 bioengineering-10-00326-f003:**
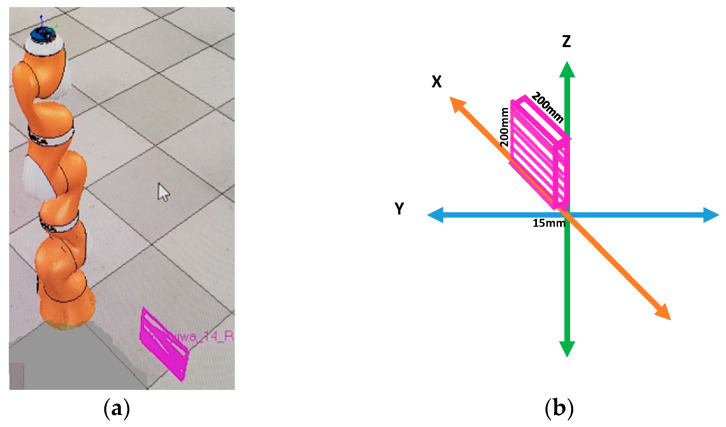
(**a**) Unwanted pHRI activities between a human participant and the Kuka robotic arm captured via V-Rep simulator, reproduced from [51], and (**b**) Cartesian representation of the 2D-XZ interactions in the 3D plane.

**Figure 4 bioengineering-10-00326-f004:**
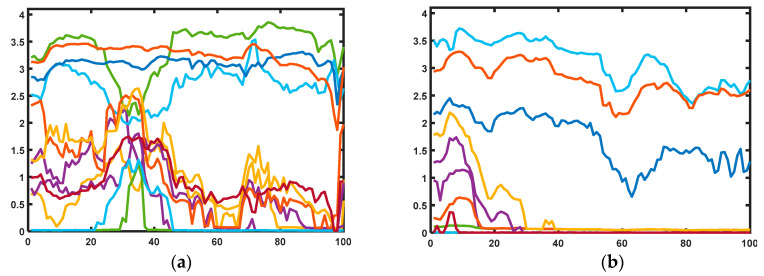
A glimpse of 16-channel FMG signals: (**a**) normal pHRI training data, (**b**) unsafe pHRI test data.

**Figure 5 bioengineering-10-00326-f005:**
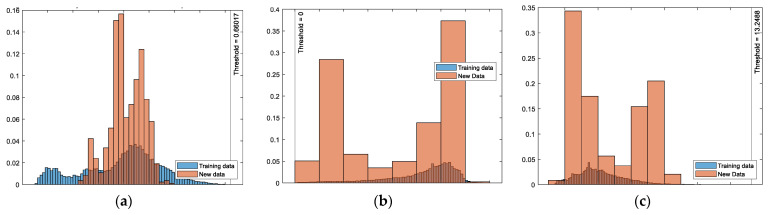
Detecting anomalies in unsafe pHRI test data in 2D-XZ plane (rep5): Histograms of novelty scores of (**a**) iForest, (**b**) OCSVM, and (**c**) Mahalanobis distance algorithms.

**Figure 6 bioengineering-10-00326-f006:**
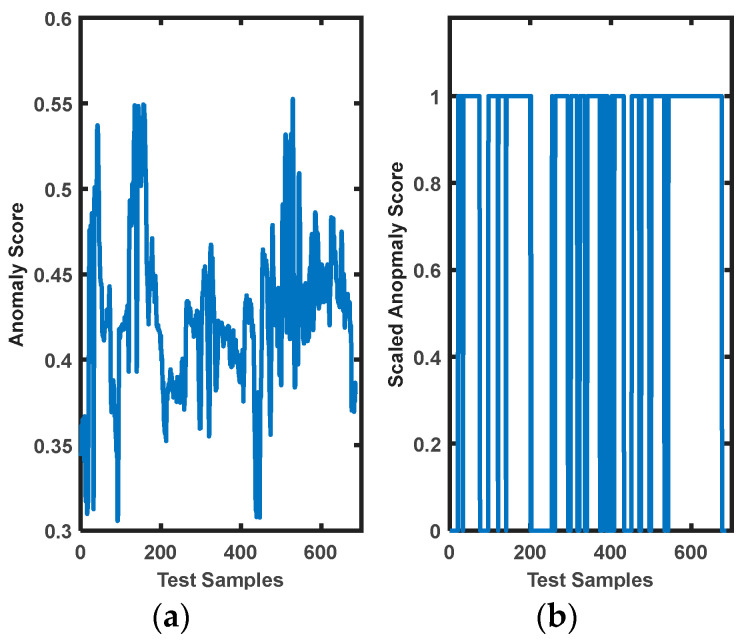
Anomaly scores of test data (rep 3) via iForest algorithm: (**a**) anomaly scores generated by the trained model, (**b**) scaled scores (scores > 0.45 were considered as novel) detecting unsafe interactions.

**Figure 7 bioengineering-10-00326-f007:**
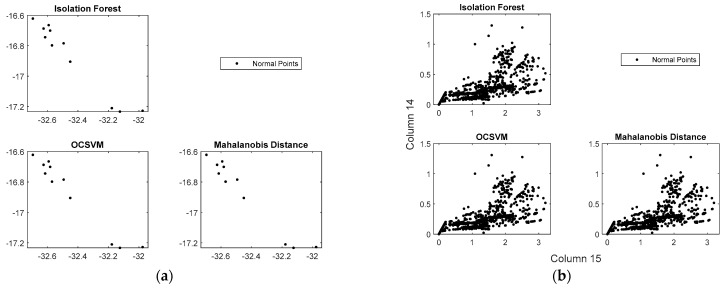
(**a**) Detecting abnormalities in the uncontaminated pHRI-ADL data, (**b**) ranking features.

**Figure 8 bioengineering-10-00326-f008:**
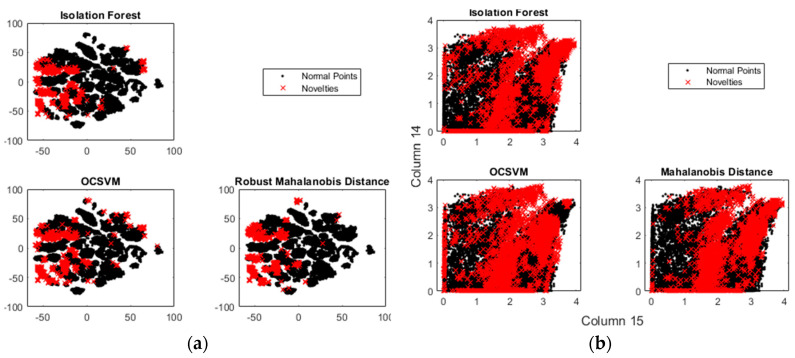
(**a**) Detecting abnormalities in the contaminated pHRI-ADL data, (**b**) ranking features.

**Table 1 bioengineering-10-00326-t001:** pHRI data.

pHRI Data	Interactions	Sample Size
Normal activities: pHRI activities of daily life (pHRI-ADL)	1D-X, Y, Z	60,743 × 16 samples
	2D-XY, YZ	
	3D-XYZ	
Abnormal activities: Anomaly or novelty detection		
	2D-XZ	686–956 × 16 samples

**Table 2 bioengineering-10-00326-t002:** Confusion matrix during evaluation.

		Predicted Anomaly Score of Test Data	
**Real anomaly score of test data**		**1**	**0**
	**1**	**TP** Test data were abnormal, and the model predicted abnormal/novel data	**FN** Test data were abnormal, but the model predicted normal data
	**0**	**FP**	**TN**

**Table 3 bioengineering-10-00326-t003:** iForest algorithm.

Training Data	Test Data		Novel Test DataDetected	Confusion Matrix	Novelty Detection	Misclassified as Normal	Elapsed Time
60,743 × 16 samples	Rep 1:	830 × 16 samples	765 × 16 samples	C = 765 65 0 0	0.9217	0.0783	1.519590 s
	Rep 2:	895 × 16 samples	855 × 16 samples	C = 855 40 0 0	0.9553	0.0447	1.062975 s
	Rep 3:	686 × 16 samples	497 × 16 samples	C = 497 189 0 0	0.7245	0.2755	1.036228 s
	Rep 4:	838 × 16 samples	784 × 16 samples	C = 784 54 0 0	0.9356	0.0644	1.046173 s
	Rep 5:	956 × 16 samples	814 × 16 samples	C = 814 142 0 0	0.8515	0.1485	1.051916 s

**Table 4 bioengineering-10-00326-t004:** OCSVM algorithm.

Training Data	Test Data		Novel Test Data Detected	Confusion Matrix	Novelty Detection	Misclassified as Normal	Elapsed Time
60,743 × 16 samples	Rep 1:	830 × 16 samples	207 × 16 samples	C = 207 623 0 0	0.2494	0.7506	1.118202 s
	Rep 2:	895 × 16 samples	104 × 16 samples	C = 104 791 0 0	0.1162	0.8838	1.102645 s
	Rep 3:	686 × 16 samples	61 × 16 samples	C = 61 625 0 0	0.0889	0.9111	0.000537 s
	Rep 4:	838 × 16 samples	82 × 16 samples	C = 82 756 0 0	0.0979	0.9021	0.001363 s
	Rep 5:	956 × 16 samples	155 × 16 samples	C = 155 801 0 0	0.1621	0.8379	1.813774 s

**Table 5 bioengineering-10-00326-t005:** Mahalanobis algorithm.

Training Data	Test Data		Novel Test Data Detected	Confusion Matrix	Novelty Detection	Misclassified as Normal	Elapsed Time
60,743 × 16 samples	Rep 1:	830 × 16 samples	830 × 16 samples	C = 830 0 0 0	1	0	1.039813 s
	Rep 2:	895 × 16 samples	895 × 16 samples	C = 895 0 0 0	1	0	1.043250 s
	Rep 3:	686 × 16 samples	686 × 16 samples	C = 686 0 0 0	1	0	1.239592 s
	Rep 4:	838 × 16 samples	838 × 16 samples	C = 838 0 0 0	1	0	1.042332 s
	Rep 5:	956 × 16 samples	956 × 16 samples	C = 956 0 0 0	1	0	1.046501 s

**Table 6 bioengineering-10-00326-t006:** Abnormality detection and reaction time via iForest algorithm (test data: rep 3).

Abnormal pHRI Test Data	Novelty Detection: Elapsed Time
Total test data: 686 × 16 samples	Accuracy in classifying anomaly = 72%
Detected as novel: 189 × 16 samples	Response time for generating scores = 3.9 s
f = 10 Hz (10 × 16 samples/s)	
T = 0.1 s = 100 ms (duration of each observation)	Classifying anomaly score as novel data and generating alert signal (elapsed time) = 1.036 s

**Table 7 bioengineering-10-00326-t007:** Data contamination.

Normal Training Data	Contaminated Training Data
Normal dataset: X_1_ 1D-X, Y, Z 2D-XY, YZ 3D-XYZ	Anomaly data: X_2_ 2D-XZ (rep 3) X dataset (X_1_ contaminated with X_2_ test data)

## Data Availability

Data can be freely downloaded at: https://doi.org/10.5281/zenodo.6632020 (accessed on 7 November 2022) under Creative Commons Attribution 4.0 International License. The corresponding author can be contacted in case of need.

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
