# Peer review of "Detecting Safety Anomalies in pHRI Activities via Force Myography"

_bioengineering, 2023, doi:10.3390/bioengineering10030326_

Round 1

Reviewer 1 Report

Authors have carried out the study well. 

They have tested three unsupervised learning algorithms to classify normal and novel/anamolous pHRI activities.

Results are well presented. 

1. In iForest study, score threshold of 0.5 is kept, based on equation 1. How is this equation taken.

What thresholds were fixed for the other algorithms and on What basis. Pl clarify. 

2.Can the authors describe some of the unwanted activities taken into consideration. 

3.what was the training testing ratio. 

4. What is the accuracy of the MD algorithm. Please form. A confusion matrix for each algorithm and study related parameters such as sensitivity and precision etc. 

5. Can this study be compared with similar studies to understand its novelty and contributions. 

Author Response

Reviewer 1 (R1)

Authors have carried out the study well.

They have tested three unsupervised learning algorithms to classify normal and novel/anamolous pHRI activities.

Results are well presented.

R1.1

  1. In iForest study, score threshold of 0.5 is kept, based on equation 1. How is this equation taken.

What thresholds were fixed for the other algorithms and on What basis. Pl clarify.

Response:

We thank the reviewer for this valuable remarks.

The iForest algorithm is based on the properties of anomalous data or outliers:

  • anomalous data are very few, so there will only be few of them in any dataset.
  • anomalous data have very different attributes than those of normal samples.

These two properties make it easier to isolate anomalous samples from the rest of the data in comparison to normal points. It tries to assign an anomaly score to observations according to the tree depth obtained recursively (binary) splitting them with a random value until an observation is alone in a branch. Since outliers will take a lesser number of splits to become isolated, they will be located closer to the root of the tree as they are easier to isolate.

The anomaly scores as defined in the original paper [33], can be in between 0 and 1. For normal samples, the score will be lower than 0.5 and closer to 0 while for anomalous samples, it will be higher than 0.5 and closer to 1. The same principle was followed for the OCSVM and the MD algorithms.

The maximum score value was determined as the default threshold value when the isolation forest was trained with the normal activities i.e., with the pHRI-ADL dataset. This default threshold was used to determine whether a test data sample was normal or anomaly or novel data. By default, the algorithm identified observations with scores above the threshold as anomalies.

[49] F. T. Liu, K. M. Ting and Z. -H. Zhou, "Isolation Forest," 2008 Eighth IEEE International Conference on Data Mining, Pisa, Italy, 2008, pp. 413-422, doi: 10.1109/ICDM.2008.17.

R1.2

Can the authors describe some of the unwanted activities taken into consideration.

Response:

We thank the reviewer for the comments.

Section 2.4. Unsafe Scenario: Target Dataset has been updated.

Action:

At Section 2.4. Unsafe Scenario: Target Dataset:

Interactions between the participant and the Kuka robot in the 2D-XZ plane was considered as anomalous data. During this unsafe and unwanted interaction, participant standing in front of the Kuka robot grasped the cylindrical gripper and applied forces in the 2D ‘XZ’ plane. In Figure 1 (a), the purple shaded box from the V-Rep simulator shows the interaction that took place. Figure 1(b) shows the pHRI interaction where the participant first pushed the robot inwards in X direction, pushed up in Z direction, and then pulled outwards in X direction and finally pulled down in Z direction.   

This 2D-XZ pHRI activity was never performed as regular activity. However, the 1D-X and 1D-Z pHRI activities had similar interactions in the single 1D plane. For example, the participant pushed the Kuka robot inwards in 1D-X about 200mm and then pulled outward in 1D-X of about 200mm. For ease of interactions, motion trajectories were confined in flexible 6-axis rectangular areas rather than fixed paths. Further description of this pHRI platform can be found in [30].

R1.3

What was the training testing ratio.

Response:

We thank the reviewer for this valuable remarks.

In the manuscript, Table 1 shows the volume of the training and the test data.

R1.4

What is the accuracy of the MD algorithm. Please form. A confusion matrix for each algorithm and study related parameters such as sensitivity and precision etc.

Response:

We thank the reviewer for this valuable suggestion.

The MD algorithms had an overall accuracy of 100% in detecting novel or unwanted data in the test scenario (for rep1-rep5).

In the manuscript, column ‘Misclassified as normal’ and column ‘Novelty detection’ Table 3, 4, and 5 shows the accuracies of the trained models in detecting test data as either normal or abnormal data. We used confusion matrix in calculating the accuracy or the volume of the training and the test data.

For clarity, we have described the equations used in accuracy calculations. Also, the confusion matrix results are included in Table 2, 3, and 4.  

Action:

Added in Section 3.1. Performance of the three algorithms

For cross-trial evaluations of the test data (rep1-rep5) via a selected algorithm, the pretrained model returned the predicted anomaly scores of the test data. For each observation of data, a score was generated, i.e., for repetition N of the test data (M X P samples), M scores between 0 and 1 were obtained. Test anomaly scores higher than 0.45 were considered as novel data (score was set to 1, indicating abnormal data) while lower scores were considered as regular data (score was set to 0, indicating normal data). A confusion matrix was generated for each test repetition, as shown in Table 2. The actual test data was considered to have scores of 1 (only abnormal data were present, i.e., negative/TN = 0, negative data/normal were absent). The predicted scores were either 1 or 0 (either abnormal or normal). The confusion matrix plotted predicted test scores either as abnormal (true positive/TP) or normal (false negative/FN) data. The set threshold helped us avoiding false positive (FP = 0) alarms and true negative (TN =0) because real test data lacked normal or negative data.

Table 2. Confusion Matrix.

                                          Predicted anomaly score

Real anomaly score

1

0

1

TP

Test data was abnormal, and the model predicted abnormal /novel data

FN

Test data was abnormal, but the model predicted normal data

0

FP

TN

For performance evaluations, accuracy in detecting novelty was calculated to find the ratio of correctly identified abnormal test data compared to the total test data. We defined accuracy in detecting novelty using the following equations:

        Acc in detecting abnormality=(TP+TN)/ (TP+FP+FN+TN)                       (3)                    

  Acc in detecting normality     =1-Acc in detecting abnormality                         (4)

The results obtained via of equation (3) and (4) are included in Table 3, 4 and 5 as the ‘Novelty detection’ and ‘Misclassified as normal.’

R1.5

Can this study be compared with similar studies to understand its novelty and contributions.

Response:

We thank the reviewer for this valuable remarks.

In section 3.5. Related Studies, Application & Future work, few studies are mentioned that focused on safety surveillance in HRI, few studies implemented unsupervised algorithms, and few studies on human biosignals. However, to our knowledge, implementing safety in HRI using the iForest, OCSVM or DM algorithms via FMG biosignals has not yet done.

Also, we have revised the contributions in Introduction section.

Action:

Added to Section: 3.5. Related Studies, Application & Future work:

In most pHRI studies, monitoring worker’s safety is based on camera surveillance. So far in our knowledge, there is no other pHRI study using FMG biosignal to detect occupational safety hazards in robotic workspace. In future, this work may be compared with related bio-signal based human-robot interaction studies.

Reviewer 2 Report

This paper studied the potential application of using a wearable force myography (FMG) band for monitoring the occupational safety of a human participant working in collaboration with an industrial robot. Three unsupervised learning algorithms: isolation forest, one-class SVM and Mahalanobis distance were trained to determine pHRI-ADL/regular activities by learning the distributions of the latent features in this paper. This research work is interesting for the force myography technique and physical human robot interactions research society. However, this paper has several limitations and the standard is not enough, and address the following item would result in a good paper,

1. The literature review is thorough and well-organized in this paper. Hence, I would like to suggest reorganizing the paper and adding more recent work in the introduction. In particular, you had better read the latest studies on force myography technique and physical human-robot interactions. Here are some relevant works that may help you to know the research advancements in this area such as (10.1109/TASE.2020.3045655)

2. In Section 2: Materials and Methods, more pictures are necessary to describe the unsupervised learning algorithms and the Physical Human Robot Interaction Platform. 

3. What are the contributions of the paper? The theoretical contribution is not enough. It suggests revising the contributions section and making these points clear and strong.

4. Comparisons should be more comprehensive, meaning that the comparison evaluation indicators should be the same for different datasets. It suggests adding comparison experiments to prove the superiority of the proposed algorithm. This comparison method could be found in these publications(10.1109/LRA.2021.3089999)(10.1109/JBHI.2019.2963048).

5. As for the proposed unwanted pHRI activity detection, it suggests enriching its application more clearly, it is far from convincing in the potential application in real applications.

6. There should be a further discussion about the limitation of the current works, in particular, what could be the challenge for its related applications. To let readers better understand future work, please give specific research directions.

Author Response

Reviewer 2 (R2)

This paper studied the potential application of using a wearable force myography (FMG) band for monitoring the occupational safety of a human participant working in collaboration with an industrial robot. Three unsupervised learning algorithms: isolation forest, one-class SVM and Mahalanobis distance were trained to determine pHRI-ADL/regular activities by learning the distributions of the latent features in this paper. This research work is interesting for the force myography technique and physical human robot interactions research society. However, this paper has several limitations and the standard is not enough, and address the following item would result in a good paper,

R2.1

  1. The literature review is thorough and well-organized in this paper. Hence, I would like to suggest reorganizing the paper and adding more recent work in the introduction. In particular, you had better read the latest studies on force myography technique and physical human-robot interactions. Here are some relevant works that may help you to know the research advancements in this area such as (10.1a109/TASE.2020.3045655)

Response:

We thank the reviewer for this comment and the suggestions.

We have revised the Introduction Section and included recent advancements in physical human-robot interactions.

Action:

In the Introduction section:

In a shared workspace, collaborative tasks with a human worker often require dynamic changes or adjustments in the predefined tasks with robots. This require implementing stable control systems for bidirectional human robot interfaces [4]. To respond to the need of effective and safe human-robot collaboration, multi-modal symbiotic communication and control methods have been used in recent years. These methods include voice commands, gesture recognition, haptic or tactile interaction, brainwave perceptions, and via machine learning and deep learning approaches to recognize awareness during interactions [1] [2][7][8]. Along with the usual surveillances, multimodal approaches of these methods are included to improve efficiency of interaction and safety.

Among these, the non-invasive surface electromyography (sEMG) signal has been in practice for decades that rely on detecting electrical current of the underlying muscle bundles in action [21]. The sEMG technique has been studied extensively for understanding human intentions in collaborative interactions during pHRI [14-20]. In multimodal environment, recognizing human arm or hand gestures via inertia measurement unit (IMU) sensors and sEMG signals along with audio and visual feedback were found effective during collaborative tasks with a collaborative robot (COBOT) [3]. The cobots are becoming increasingly popular in shared workspace because these are de-signed to work with human workers with the advanced hand-guiding techniques based on their built—in torque sensors in every joints.

R2.2

In Section 2: Materials and Methods, more pictures are necessary to describe the unsupervised learning algorithms and the Physical Human Robot Interaction Platform. 

Response:

We thank the reviewer for the valuable comment. The Section 2: Materials and Methods is modified and revised, restructured y adding subsections from Section 3: Results for better readability and appropriate flow of the work.

Action:

Added the following figure in Section 1:

Figure 1. The proposed FMG-based occupational health & safety monitoring system.

R2.3

What are the contributions of the paper? The theoretical contribution is not enough. It suggests revising the contributions section and making these points clear and strong.

Response:

We thank the reviewer for the suggestion.

We have revised and reorganized the manuscript clarifying the contribution in the Section 1: Introduction and discussed limitation of the proposed system in a new subsection 3.4. Discussions & Limitations in the Section 3: Results.

Action:

In Section 1: Introduction

The main contributions of this study were:

  • defining normal, daily interactions between a human worker and a robot that were preset and carried out regularly as ‘pHRI-ADL’ activities via FMG-based complaint collaboration,
  • implementing unsupervised learning techniques to understand the daily human robot interactive tasks via iForest, OCSVM and MD algorithms by training models with the normal pHRI-ADL distributions,
  • evaluating the trained models in detecting unsafe human-robot collaborative interactions via FMG biosignals that were novel, inlier, or outlier to the normal distribution,
  • generating warnings when unsafe or hazardous interactions were detected,
  • feasibility of the proposed system to react quickly in detecting novel, unsafe interactions and generating alerts to prevent injuries, and hence,
  • providing an extra layer of occupational safety feature in addition to the complaint collaboration in human-robot workplaces.

In this Industry 4.0 era, automobile industries are swiftly embracing technologies such as IoT, cloud computing, artificial intelligence, and big data analytics to achieve in-creasing productivity. For collaborative HRI tasks in this setting, the use of FMG armbands can support controlled and safe interactions. Hence, this study proposed the viability of using FMG biofeedback to provide additional safety layer along with complaint collaboration in human-robot workplaces that require human participations via Industry 4.0 technologies.

R2.4

Comparisons should be more comprehensive, meaning that the comparison evaluation indicators should be the same for different datasets. It suggests adding comparison experiments to prove the superiority of the proposed algorithm. This comparison method could be found in these publications (10.1109/LRA.2021.3089999) (10.1109/JBHI.2019.2963048).

Response:

We thank the reviewer for this valuable comment and the reference.

We have revised the title of section 3.3. Comparing Algorithms in Abnormal pHRI Activity Detection as 3.3. Normal Vs Abnormal or unsafe pHRI Activity Detection.

The three trained models were evaluated separately in the test scenario, classification results were summarized as confusion matrix and corresponding accuracies in detecting novel (unsafe) data were calculated. These results are shown in Table 2,3, and 4 for individual model.

Action:

Table 2,3, and 4 are updated as 3,4, and 5.

Added to Section 3.1. Performance of the three algorithms

For cross-trial evaluations of the test data (rep1-rep5) via a selected algorithm, the pretrained model returned the predicted anomaly scores of the test data. For each observation of data, a score was generated, i.e., for repetition N of the test data (M X P samples), M scores between 0 and 1 were obtained. Test anomaly scores higher than 0.45 were considered as novel data (score was set to 1, indicating abnormal data) while lower scores were considered as regular data (score was set to 0, indicating normal data). A confusion matrix was generated for each test repetition, as shown in Table 2. The actual test data was considered to have scores of 1 (only abnormal data were present, i.e., negative/TN = 0, negative data/normal were absent). The predicted scores were either 1 or 0 (either abnormal or normal). The confusion matrix plotted predicted test scores either as abnormal (true positive/TP) or normal (false negative/FN) data. The set threshold helped us avoiding false positive (FP = 0) alarms and true negative (TN =0) because real test data lacked normal or negative data.

Table 2. Confusion Matrix.

                                          Predicted anomaly score

Real anomaly score

1

0

1

TP

Test data was abnormal, and the model predicted abnormal /novel data

FN

Test data was abnormal, but the model predicted normal data

0

FP

TN

For performance evaluations, accuracy in detecting novelty was calculated to find the ratio of correctly identified abnormal test data compared to the total test data. We defined accuracy in detecting novelty using the following equations:

        Acc in detecting abnormality=(TP+TN)/ (TP+FP+FN+TN)                       (3)                    

  Acc in detecting normality     =1-Acc in detecting abnormality                         (4)

The results obtained via of equation (3) and (4) are included in Table 3, 4 and 5 as the ‘Novelty detection’ and ‘Misclassified as normal.’

Added a new Section 3.4: Discussions & Limitations

To evaluate the performances of the classifiers in detecting anomalies, we used anomaly score, histogram-based novelty score and Laplacian score with same baseline parameters and thresholds for consistencies. Such as, for all three algorithms, anomaly score of values less than 0.5 were considered as normal and values above 0.5 were anomalies. Detecting patterns or anomalies was plotted and visualized in histogram-based novelty scores. The training distributions and test data points in the plotted histograms showed quick identification of outliers, patterns or trends based on the degree of anomaly calculations. The Laplacian score was computed for each feature based on the locality preserving property (data from the same class were often close to each other). It selected variables with the smallest scores and ranked features of the observations using the Laplacian scores.

To evaluate the classifiers, we trained all three models with the same training datasets that were collected during the preset, daily human-robot interactions. Each training interaction in the 1D-X, Y, Z plane, 2D-XY, YZ plane and 3D-XYZ plane were different and hence, collected data were different in terms of applied interactive forces and motion trajectories. Even in a selected interaction such as in 1D-X plane, the 5 repetitions of collected data were not exactly same because of unpredictable human behavior where interactions were confined in a certain 6-axis rectangular area rather than in a fixed motion path. These pretrained models were evaluated on test data where unwanted interactions happened between the human and the robot. The anomaly score, histogram-based novelty score and the Laplacian score were calculated for each model. For fair comparison, cross-trial evaluations of detecting unwanted interactions were carried out several times. It is worth to acknowledge that in the 5 repetitions of the test interactions, same unwanted event happened, but these data were not exactly same. Because each trial/repetition was unique due to the dynamic nature of human robot interactions in the 2D plane where the motion trajectory was not fixed between a start and end point, rather it was bounded by a 6-axis rectangular area in the 2D space. So, the cross-trial evaluations showed the viability of the proposed system even for a single test scenario. 

As this study was done in a strictly monitored experiment environment with certain setup, we defined few regular, preset tasks and one unwanted test scenario where a human participant interacted with a 7-DoF serial manipulator. The models were trained with different interactions in the 1D 2D and 3D planes which were separate from each other. It was done with the intention to represent a variety of pre-defined industrial human robot interactive tasks as regular tasks. In the test scenario, we selected an interaction that was unwanted, undefined, and impulsive suddenly occurring between the human and the robot. The models were examined separately for 5 times as cross-trial evaluations of the test scenario. It was observed that the models were able to detect anomalies or novel, unsafe interactions and could alert occurrence of a potential health and safety hazard within a second not only for the best case (100% anomaly detection via MD model) but also for the worst case (9% of anomaly detection via OCSVM model). Hence, even the proposed system was found viable with the experimental training and test applications, we believe it could feasibly be applied in industrial scenarios of real human-robot interactive applications. Although further studies involving more human participants are required in future.

R2.5

As for the proposed unwanted pHRI activity detection, it suggests enriching its application more clearly, it is far from convincing in the potential application in real applications.

Response:

We are grateful to the reviewer for the comment and agree that our training or test scenarios were different from real applications.

As this study was done in an experiment environment with certain setup, we defined few regular, preset tasks and one unwanted test scenario where a human participant interacted with a 7-DoF serial manipulator. The models were trained with different interactions in the 1D 2D and 3D planes which were completely separate from each other. It was done with the intention to represent a variety of pre-defined industrial human robot interactive tasks as regular tasks. In the test scenario, we selected an interaction that was unwanted, undefined, and impulsive suddenly occurring between the human and the robot. The models were examined separately for 5 times as cross-trial evaluations of the test scenario. It was observed that the models were able to detect anomalies or novel, unsafe interactions and could alert occurrence of a potential health and safety hazard within a second not only for the best case (100% anomaly detection via MD model) but also for the worst case (9% of anomaly detection via OCSVM model). Hence, even the proposed system was found viable with the experimental training and test applications, we believe it could feasibly be applied in industrial scenarios of real human-robot interactive applications. Although further studies are required in future.

Action:

Added at Section 3.4:

As this study was done in a strictly monitored experiment environment with certain setup, we defined few regular, preset tasks and one unwanted test scenario where a human participant interacted with a 7-DoF serial manipulator. The models were trained with different interactions in the 1D 2D and 3D planes which were separate from each other. It was done with the intention to represent a variety of pre-defined industrial human robot interactive tasks as regular tasks. In the test scenario, we selected an interaction that was unwanted, undefined, and impulsive suddenly occurring between the human and the robot. The models were examined separately for 5 times as cross-trial evaluations of the test scenario. It was observed that the models were able to detect anomalies or novel, unsafe interactions and could alert occurrence of a potential health and safety hazard within a second not only for the best case (100% anomaly detection via MD model) but also for the worst case (9% of anomaly detection via OCSVM model). Hence, even the proposed system was found viable with the experimental training and test applications, we believe it could feasibly be applied in industrial scenarios of real human-robot interactive applications. Although further studies involving more human participants are required in future.

R2.6

There should be a further discussion about the limitation of the current works, in particular, what could be the challenge for its related applications. To let readers better understand future work, please give specific research directions.

Response:

We are grateful to the reviewer for the comment and the suggestion.

As this study was done in a strictly monitored experiment environment with certain setup, we defined few regular, preset tasks and one unwanted test scenario where a human participant interacted with a 7-DoF serial manipulator. The proposed system is required to be evaluated in real industrial human-robot collaborative tasks where more uncertainties are involved. Also, involving more participants would also justify the proposal.

Future research direction was provided in Section 3.5. Related Studies, Application & Future work:

In the future, FMG biosignals can be used with other sensory devices such as IMU data or with vision system for a comprehensive workspace safety design. Also, the approaches adopted here can be further studied in other fields where understanding human activities can be insightful.

Action:

Section 3.4 is added on discussions and limitations of the proposed study while section 3.5 is modified and revised.

Reviewer 3 Report

The manuscript presented an interesting research topic on FMG bio-feedback to ensure occupational safety in robotic workplaces. Three unsupervised learning algorithms (iForest algorithm; One-Class Learning or unsupervised SVM (OCSVM) algorithm; and Mahalanobis Distance (MD) algorithm) are proposed to unwanted pHRI activity detection.

Abstract

The abstract is a concise description of the work.

Introduction

The introduction is well structured, and it covers all the concepts investigated in the methodological part. The literature review is well presented and integrated but could be improved with more discussion and valuable references, as shown below.

Valuable references and small discussion about technologies and methodologies used for interaction and safety were missed between lines 45 and 47, such as:

https://www.doi.org/10.3390/robotics8040100

https://www.doi.org/10.1108/SR-10-2014-0723

https://www.doi.org/10.1007/978-3-319-27149-1_8

https://www.doi.org/10.1016/j.cirp.2019.05.002

In line 54, you are missing a short discussion of the most common form of physical human-robot interaction, i.e., using physical contact, force/torque sensors. Important references on this topic are:

https://www.doi.org/10.1007/s10514-017-9677-2

https://www.doi.org/10.1109/INDIN.2018.8472058

https://www.doi.org/10.1016/j.robot.2010.07.002

Valuable references about sEMG-based HRI were missed between lines 61 and 67, such as:

https://www.doi.org/10.3390/s18051615

https://www.doi.org/10.1016/j.promfg.2017.07.158

https://www.doi.org/10.1109/THMS.2018.2883176

https://www.doi.org/10.1109/ACCESS.2019.2906584

https://www.doi.org/10.1109/tbme.2019.2899222

https://www.doi.org/10.1109/IECON.2019.8927221

https://www.doi.org/10.1007/s10846-022-01666-5

https://www.doi.org/10.1109/TNSRE.2013.2247631

I consider that this work brings added value in the field and the specific objectives of the manuscript are well related to the previous work developed in this domain.

Methodology

The research design used is appropriate to answer the research questions proposed by the authors. However, more detail about the FMG data processing and the implementation of the three algorithms (iForest algorithm; One-Class Learning or unsupervised SVM (OCSVM) algorithm; and Mahalanobis Distance (MD) algorithm) for detecting unsafe pHRI activities should be presented. The presentation of schematics would help in understanding the methodology. More details about the FMG band, as well as an illustrative schematic and a better-quality real image of the device should be provided. The results are clearly presented and are in relation to the concepts investigated. However, the first column of Tables 2, 3 and 4 is not understandable.

Discussion and conclusions

The discussions are clear and concise. The conclusions are strongly related to the findings of the research work.

Format and style

All the format and style features were respected and are compliant with the requirements.

References

The format of the reference list is suitable for the specified format. However, there are minor typos in some references, for example ref 32, 34, 35, etc.

Figures

Figure 1 shows low quality. The FMG device should be shown in more detail, in schematic or real image.

Acronyms

The acronym pHRI should be defined in lines 14 and 57 and then used again and again throughout the document, change line 125.

Author Response

Reviewer 3 (R3)

The manuscript presented an interesting research topic on FMG bio-feedback to ensure occupational safety in robotic workplaces. Three unsupervised learning algorithms (iForest algorithm; One-Class Learning or unsupervised SVM (OCSVM) algorithm; and Mahalanobis Distance (MD) algorithm) are proposed to unwanted pHRI activity detection.

R3.1

Abstract

The abstract is a concise description of the work.

Response:

We thank the reviewer for this comment.

R3.2

Introduction

The introduction is well structured, and it covers all the concepts investigated in the methodological part.

Response:

We thank the reviewer for the comment.

R3.3

The literature review is well presented and integrated but could be improved with more discussion and valuable references, as shown below.

Valuable references and small discussion about technologies and methodologies used for interaction and safety were missed between lines 45 and 47, such as:

https://www.doi.org/10.3390/robotics8040100

https://www.doi.org/10.1108/SR-10-2014-0723

https://www.doi.org/10.1007/978-3-319-27149-1_8

https://www.doi.org/10.1016/j.cirp.2019.05.002

In line 54, you are missing a short discussion of the most common form of physical human-robot interaction, i.e., using physical contact, force/torque sensors. Important references on this topic are:

https://www.doi.org/10.1007/s10514-017-9677-2

https://www.doi.org/10.1109/INDIN.2018.8472058

https://www.doi.org/10.1016/j.robot.2010.07.002

Valuable references about sEMG-based HRI were missed between lines 61 and 67, such as:

https://www.doi.org/10.3390/s18051615

https://www.doi.org/10.1016/j.promfg.2017.07.158

https://www.doi.org/10.1109/THMS.2018.2883176

https://www.doi.org/10.1109/ACCESS.2019.2906584

https://www.doi.org/10.1109/tbme.2019.2899222

https://www.doi.org/10.1109/IECON.2019.8927221

https://www.doi.org/10.1007/s10846-022-01666-5

https://www.doi.org/10.1109/TNSRE.2013.2247631

I consider that this work brings added value in the field and the specific objectives of the manuscript are well related to the previous work developed in this domain.

Response:

We thank the reviewer for the valuable comment and suggestions.

We have revised the Introduction section and included the new references as suggested.

Action:

Included in Introduction:

In a shared workspace, collaborative tasks with a human worker often require dynamic changes or adjustments in the predefined tasks with robots. This require implementing stable control systems for bidirectional human robot interfaces [4].  To respond to the need of effective and safe human-robot collaboration, multi-modal symbiotic communication and control methods have been used in recent years. These methods include voice commands, gesture recognition, haptic or tactile interaction, brainwave perceptions, and via machine learning and deep learning approaches to recognize awareness during interactions [1][2][7][8]. Along with the usual surveillances, multimodal approaches of these methods are included to improve efficiency of interaction and safety.

In multimodal environment, recognizing human arm or hand gestures via inertia measurement unit (IMU) sensors and sEMG signals along with audio and visual feedback were found effective during collaborative tasks with a collaborative robot (cobot) [3]. The cobots are becoming increasingly popular in shared workspace because these are de-signed to work with human workers with the advanced hand-guiding techniques based on their built—in torque sensors in every joints.

We have rewritten the contributions.

The main contributions of this study were:

─             The main contributions of this study were:

─             defining normal, daily interactions between a human worker and a robot that were preset and carried out regularly as ‘pHRI-ADL’ activities via FMG-based complaint collaboration,

─             implementing unsupervised learning techniques to understand the daily human robot interactive tasks via iForest, OCSVM and MD algorithms by training models with the normal pHRI-ADL distributions,

─             evaluating the trained models in detecting unsafe human-robot collaborative interactions via FMG biosignals that were novel, inlier, or outlier to the normal distribution,

─             generating warnings when unsafe or hazardous interactions were detected,

─             feasibility of the proposed system to react quickly in detecting novel, unsafe interactions and generating alerts to prevent injuries, and hence,

─             providing an extra layer of occupational safety feature in addition to the complaint collaboration in human-robot workplaces.

R3.4

Methodology

The research design used is appropriate to answer the research questions proposed by the authors. However, more detail about the FMG data processing and the implementation of the three algorithms (iForest algorithm; One-Class Learning or unsupervised SVM (OCSVM) algorithm; and Mahalanobis Distance (MD) algorithm) for detecting unsafe pHRI activities should be presented. The presentation of schematics would help in understanding the methodology. More details about the FMG band, as well as an illustrative schematic and a better-quality real image of the device should be provided. The results are clearly presented and are in relation to the concepts investigated. However, the first column of Tables 2, 3 and 4 is not understandable.

Response:

We thank the reviewer for these suggestions.

We have revised Section 2.3 and 2.4 to add details on the HRI platform and the FMG data during interactions. Section 2.4 has detailed FMG data collection and processing.

We have changed Section 3.1 as Section 2.5 Implementing Novelty Detection via Unsupervised ML Algorithms which already covered the implementations of the 3 selected algorithms and Section and 3.2 as Section 2.6 Performance Analysis Tools that described how the algorithms were evaluated.

We have included a diagram of the graphical abstract for clear presentation of the work and better-quality images. 

The first column of Tables 2, 3 and 4 is the volume of the data collected during preset, normal activities defined as the pHRI-ADL in the1D-X, -Y, -Z, 2D-XY, -YZ and 3D-XYZ planes.

Action:

Included in Materials and Methods:

2.3 Physical Human Robot Interaction Platform

The forearm FMG band had 16 FSRs (TPE 502C, Tangio Printed Electronics, Vancouver, Canada), was roughly 30 cm long and connected to an external PC (Intel Core i7 processor and Nvidia GTX-1080 GPU) via a data acquisition device (NI USB 6341, National Instruments, Austin, TX, US) to collect data at ~50 Hz.

Our target environment was the industrial manufacturing assembly line where huge mechanical robots were engaged. These robots did not have any torque sensors in their joints to recognize externally applied forces by human workers during interactions.  Replacing these robots or including joint torque sensors for safe human robot collaboration were neither cost-effective nor feasible. As we needed to investigate improved safety and surveillance techniques for such a workspace, we monitored human interactions with a Kuka robot in this research. The manipulator had an 820mm reach with a 14 Kg payload and built-in torque sensors in all joints except the end-effector. But we did not utilize these torque sensors so that we could replicate the industrial robots.

2.4. pHRI Data Collection

The pHRI datasets were collected during dynamic interactions between a participant and the Kuka robot in 1D, 2D and 3D space. The participant grasped the cylindrical gripper in a standing position in front of the robot and applied forces in dynamic motions wearing a 16 channel FMG forearm band on his dominant right arm, as shown in Figure 1 (c). Data for several interactions were collected in the 1D, 2D and 3D planes. For all interactions, 5 repetitions of sample data (applied human forces in dynamic motions during interactions with the robot) were collected. All data were balanced, and no missing values were present in the datasets. So preprocessing was not required to reduce memory consumption and speed up training. Raw FMG signals were used as the predictor data, organized as CSV files. Each row corresponded to one observation while each column corresponded to one predictor variable. Hence, for 16-channel FMG band, 16 features or predictor variables were present. Each interaction continued for about 60-90 seconds, termed as one ‘repetition’, and contributed approximately 600-900 X 16 samples at a ~10-12 Hz sampling rate. A detailed description of this dataset can be found in [32]. Table 1 shows the normal and abnormal dataset that were used in this study for pHRI anomaly detection.

  • Normal Scenario: pHRI-ADL Training Dataset

The preplanned human robot interactions that happened between the human participant and the Kuka robot as regular tasks were termed as the activities of daily life in pHRI (pHRI-ADL). Such as, interactions between the participant and the kuka robot in 1D (X, Y, and Z directions), 2D (XY and YZ planes) and 3D (XYZ plane) were considered as pHRI-ADL activities. In each task (interactions in 1D-X), human applied external force in a distinguished direction, and the robot’s trajectory became different than other interactions or tasks. Five (5) repetitions of similar FMG data were collected for one task or interaction. Hence, a total of 60,743 X 16 FMG samples from 30 repetitions (1D plane: 15 reps, 2D plane: 10 reps, 3D plane: 5 reps) were collected. These variety of FMG signals from the pHRI-ADL activities were used as normal training dataset to train three separate unsupervised models via the three selected algorithms. 

R3.5

Discussion and conclusions

The discussions are clear and concise. The conclusions are strongly related to the findings of the research work.

 Response:

We thank the reviewer for this nice comment.

R3.6

Format and style

All the format and style features were respected and are compliant with the requirements.

Response:

We thank the reviewer for the comment.

R3.7

References

The format of the reference list is suitable for the specified format. However, there are minor typos in some references, for example ref 32, 34, 35, etc.

Response:

We thank the reviewer for the comment and have revised the Reference section with new additions as suggested.

R3.8

Figures

Figure 1 shows low quality. The FMG device should be shown in more detail, in schematic or real image.

Response:

We thank the reviewer for the comment and have updated Figure 1.

R3.9

Acronyms

The acronym pHRI should be defined in lines 14 and 57 and then used again and again throughout the document, change line 125.

Response:

We thank the reviewer for the comment and have revised accordingly.

Round 2

Reviewer 2 Report

The study explored the potential of using a wearable force myography (FMG) band for monitoring human-robot collaboration safety in industrial settings. Data-driven models were used to detect abnormal movements and generate warning signals to prevent injuries and improve occupational health. The results demonstrated the feasibility of using FMG biofeedback for safety monitoring in workplaces that require human-robot collaboration.

Scope: Although using FMG biofeedback to monitor occupational safety is an interesting concept, the study needs to present significant innovative ideas or approaches beyond using data-driven models and unsupervised learning algorithms to detect unusual movements.

Review: The paper is built on a thorough and well-organized literature review. Hence, I would like to suggest reading the latest studies and adding a new section behind the introduction to enrich further.

Figures: Some figures are of low resolution and should be adjusted to be more improved and readable. Such as fig.1 and fig.3. 

Novelty: Although using FMG biofeedback to monitor occupational safety is an interesting concept, the study needs to present significant innovative ideas or approaches beyond using data-driven models and unsupervised learning algorithms to detect unusual movements.

Methodology: The study relies on unsupervised learning algorithms to detect unusual movements, which may need to be sufficiently accurate or reliable in all situations. The validity and limitations of the proposed approach should be further discussed.

Discussion: The paper only briefly discusses other research on the use of FMG biofeedback for monitoring occupational safety, without a comprehensive evaluation or discussion of how this study relates to and extends prior work.

Language or grammar: Language expression needs further improvement to make the article more readable. It is recommended to double-check the spelling of words and sentence grammar.

Author Response

Thank you for the valuable comments and suggestions.

Reviewer 2 (R2)

The study explored the potential of using a wearable force myography (FMG) band for monitoring human-robot collaboration safety in industrial settings. Data-driven models were used to detect abnormal movements and generate warning signals to prevent injuries and improve occupational health. The results demonstrated the feasibility of using FMG biofeedback for safety monitoring in workplaces that require human-robot collaboration.

R2.1

Scope: Although using FMG biofeedback to monitor occupational safety is an interesting concept, the study needs to present significant innovative ideas or approaches beyond using data-driven models and unsupervised learning algorithms to detect unusual movements.

Response:

We thank the reviewer for the valuable comment.

In this preliminary study, our focus was to investigate the viability of a single biosensory data for safety monitoring along with compliant collaboration via admittance control. The FMG signals provided useful information to interpret human motions and to detect possible threat or hazardous situation via data-driven approaches. Adopting data-driven approaches were found advantageous in facilitating collaboration by learning motion patterns to recognize human feedback. These approaches also helped to avoid complex and time-consuming dynamic arm modelling of individual human participant. The unsupervised learning techniques used in this study showed the viability of detecting normal or abnormal movements during human robot interactions and generated alert signal for manual or automated override. Further improvements can be achieved by developing an automated decision making system with collision-avoidance trajectory planning for the robot once unsafe interactions is identified.  

This study was done in a strictly monitored experiment environment with a certain setup with one human participant wearing a forearm FMG band. Future studies can include another FMG band in upper arm to record shoulder abduction and adduction, an IMU sensor to track arm pose and orientation and a vision system to capture unsafe interactions. Multimodal sensory data would be more reliable in understanding human intentions and monitoring potential safety threats during interactions in industrial workspaces.

Action:

We have updated the Section 4.2. Related Studies, Application & Future work.

Added in the Section 4.2.:

Adopting data-driven approaches were found advantageous in facilitating collaboration by learning motion patterns to recognize human feedback. These approaches also helped to avoid complex and time-consuming dynamic arm modelling of individual human participant. The unsupervised learning techniques used in this study showed the viability of detecting normal or abnormal movements during human robot interactions and generated alert signal for manual override. Further improvements can be achieved by developing an automated decision making system with collision-avoidance trajectory planning for the robot once unsafe interactions is identified.

In this preliminary study, our focus was to investigate the viability of a single bio-sensory data for safety monitoring along with compliant collaboration via admittance control. The FMG signals provided useful information to interpret human motions and to detect possible threat or hazardous situation. The study was conducted in a strictly monitored experiment environment where one human participant wearing a forearm FMG band interacted with a serial manipulator. Future studies can include another FMG band in upper arm to record shoulder abduction and adduction, an IMU sensor to track arm pose and orientation and a vision system to capture unsafe interactions. Multimodal sensory data would be more reliable in understanding human intentions and monitoring potential safety threats during interactions in industrial workspaces.

R2.2

Review: The paper is built on a thorough and well-organized literature review. Hence, I would like to suggest reading the latest studies and adding a new section behind the introduction to enrich further.

Response:

We thank the reviewer for this comment and the suggestion.

We have revised the Introduction Section and included recent advancements in physical human-robot interactions.

Action:

In the Introduction section:

Usually, worker’s safety in industrial workspaces with robots means physical safety which is mostly associated with collision risks.  Hence, in most cases, maintaining a safe distance with robots would be appropriate to avoid collisions. Recent research was con-ducted using a practical vision-based human safety system via mixed reality (MR) for establishing a minimum safe distance [19]. Ensuring safety and optimizing performance of HRI task was investigated as a stochastic optimal control problem (OCP) that included human motions. For collaborative tasks, adapting human behavior using camera and op-tical tracker was taken into consideration for collision-free trajectory planning [20].  Realtime point-to-point motion planning in shared workspace via non-linear predictive model was studied in [21]. Authors in [22] investigated recognizing human pose via sensors for context-awareness-based collision-free human-robot collaboration (HRC). Recent approaches used a supervisory control scheme often known as ‘shielding’ for safe robot operation in uncertain environments. Authors in [23] relied on predicting future shielding events that could override robot’s nominal plan with a safety fallback strategy in critical situations. In this Industry 4.0 era, many of these studies used several monitoring devices and relied on deep learning approaches, IoT, cloud computing, mixed reality for real-time control and safety. However, these studies focused on maintaining safe distances rather than direct contact between a human worker and a robot.   

R2.3

Figures: Some figures are of low resolution and should be adjusted to be more improved and readable. Such as fig.1 and fig.3. 

Response:

We thank the reviewer for this comment and the suggestion.

We have included images with higher resolutions.

R2.4

Novelty: Although using FMG biofeedback to monitor occupational safety is an interesting concept, the study needs to present significant innovative ideas or approaches beyond using data-driven models and unsupervised learning algorithms to detect unusual movements.

Response:

We thank the reviewer for this comment.

We have updated the Section 4.2. Related Studies, Application & Future work.

Action:

Added in the Section 4.2.:

Adopting data-driven approaches were found advantageous in facilitating collaboration by learning motion patterns to recognize human feedback. These approaches also helped to avoid complex and time-consuming dynamic arm modelling of individual human participant. The unsupervised learning techniques used in this study showed the viability of detecting normal or abnormal movements during human robot interactions and generated alert signal for manual override. Further improvements can be achieved by developing an automated decision making system with collision-avoidance trajectory planning for the robot once unsafe interactions is identified.

In this preliminary study, our focus was to investigate the viability of a single bio-sensory data for safety monitoring along with compliant collaboration via admittance control. The FMG signals provided useful information to interpret human motions and to detect possible threat or hazardous situation. This study was performed in a strictly monitored experiment environment with one human participant wearing a forearm FMG band. Future studies can include another FMG band in upper arm to record shoulder abduction and adduction, an IMU sensor to track arm pose and orientation and a vision system to capture unsafe interactions. Multimodal sensory data would be more reliable in understanding human intentions and monitoring potential safety threats during interactions in industrial workspaces.

R2.5

Methodology: The study relies on unsupervised learning algorithms to detect unusual movements, which may need to be sufficiently accurate or reliable in all situations. The validity and limitations of the proposed approach should be further discussed.

Response:

We thank the reviewer for this comment and the suggestions.

We have updated the 4.1. Discussions & Limitations Section and included validity and limitations of the proposed approach.

Action:

Added in the 4.1. Discussions & Limitations Section:

This study relied on data-driven models to learn latent features of FMG biosignals in detecting anomalies or safety hazards. The trained model used a variety of pHRI-ADL or regular data to detect unusual or irregular pHRI data. During evaluation, detecting novel or unwanted interactions was possible because the hazardous movement was significantly different than the usual tasks. Hence, for real-time practical applications, collection of every possible regular activity data is required to ensure safety in industrial workspaces. In a situation when safety hazards arise during usual interactions and the worker tries to continue the preset fixed task, it might become hard for the data-driven model to detect anomalies. The model might not succeed to determine safety hazards if the unwanted interaction becomes similar to the regular data. Therefore, the proposed safety detection in this study can be implemented as an additional feature with the usual surveillance monitoring that prevail in industry.    

R2.6

Discussion: The paper only briefly discusses other research on the use of FMG biofeedback for monitoring occupational safety, without a comprehensive evaluation or discussion of how this study relates to and extends prior work.

Response:

We thank the reviewer for this remarkable comment.

We have updated the 4.1. Discussions & Limitations Section.

Action:

Added in the 4.1. Discussions & Limitations Section:

In contrast to multimodal safety systems, we relied on one bio-sensory data i.e., the FMG data -for both the underlying compliant collaboration control layer and the proposed safety monitoring as an additional layer. This study was an extension of the previous works where instantaneous applied hand forces were predicted for admittance control via a supervised convolutional neural network model [46]. The focus was to investigate safety in collaborative tasks when the human worker had direct contact with the robot rather than implementing a collision-free or collision-avoidance system. With the FMG-based admittance control, a human worker could manually push away the robot if needed, but it was not possible to know occurrences of potential safety hazards. This motivated us to build additional safety layer for unwanted pHRI detection using the same bio-sensory data. This helped in reducing control signal communication and computational complexity by having one sensory data instead of multimodal data. Hence, the proposed system can be used as an only source sensory data in combination with prominent surveillances systems such as the vision system. It can provide additional layer of protection within the existing safety protocol for human-robot workspaces rather than as a stand-alone safety system.

R2.7

Language or grammar: Language expression needs further improvement to make the article more readable. It is recommended to double-check the spelling of words and sentence grammar.

Response:

We thank the reviewer for this comment and the suggestions.

We have revised the manuscript for grammatical errors and typos.
